# Influence of Long-Term Anti-Seizure Medications on Redox Parameters in Human Blood

**DOI:** 10.3390/ph17010130

**Published:** 2024-01-18

**Authors:** Danijel Jakovljević, Milan Nikolić, Vesna Jovanović, Teodora Vidonja Uzelac, Aleksandra Nikolić-Kokić, Emilija Novaković, Čedo Miljević, Maja Milovanović, Duško Blagojević

**Affiliations:** 1Department of Biochemistry, Faculty of Chemistry, University of Belgrade, 11158 Belgrade, Serbia; jakovljevicdanijel96@gmail.com (D.J.); mnikolic.chem@gmail.com (M.N.); vjovanovic@chem.bg.ac.rs (V.J.); 2Department of Physiology, Institute for Biological Research “Siniša Stanković”, National Institute of Republic of Serbia, University of Belgrade, 11108 Belgrade, Serbia; teodora.vidonja@ibiss.bg.ac.rs (T.V.U.); dblagoje@ibiss.bg.ac.rs (D.B.); 3Clinic for Mental Disorders “Dr. Laza Lazarević”, 11000 Belgrade, Serbia; emilija.novakovic@med.pr.ac.rs; 4Faculty of Medicine, University of Priština, 38220 Kosovska Mitrovica, Serbia; 5Outpatient Department, Institute of Mental Health, School of Medicine, University of Belgrade, 11000 Belgrade, Serbia; cedo.miljevic35@gmail.com; 6Department for Epilepsy and Clinical Neurophysiology, Institute of Mental Health, Faculty for Special Education and Rehabilitation, University of Belgrade, 11000 Belgrade, Serbia; jmtmilov@gmail.com

**Keywords:** anti-seizure medications, redox homeostasis, antioxidant enzymes, erythrocytes, plasma

## Abstract

Background: Epilepsy is a chronic brain disease affecting millions of people worldwide, but little is known about the impact of anti-seizure medications on redox homeostasis. Methods: This study aimed to compare the effects of the long-term use of oral anti-seizure medications in monotherapy (lamotrigine, carbamazepine, and valproate) on antioxidant enzymes: superoxide dismutase, catalase, glutathione peroxidase, glutathione reductase, haemoglobin, and methaemoglobin content in erythrocytes, and concentrations of total proteins and thiols, nitrites, lipid peroxides and total glutathione in the plasma of epilepsy patients and drug-naïve patients. Results: The results showed that lamotrigine therapy led to lower superoxide dismutase activity (*p* < 0.005) and lower concentrations of total thiols (*p* < 0.01) and lipid peroxides (*p* < 0.01) compared to controls. On the other hand, therapy with carbamazepine increased nitrite levels (*p* < 0.01) but reduced superoxide dismutase activity (*p* < 0.005). In the valproate group, only a decrease in catalase activity was observed (*p* < 0.005). Canonical discriminant analysis showed that the composition of antioxidant enzymes in erythrocytes was different for both the lamotrigine and carbamazepine groups, while the controls were separated from all others. Conclusions: Monotherapy with anti-seizure medications discretely alters redox homeostasis, followed by distinct relationships between antioxidant components.

## 1. Introduction

Epilepsy is one of the most common neurological disorders in children and adults. It is described as a group of disorders characterised by chronic, recurrent, and paroxysmal changes in motor and sensory neurological functions due to a disturbance in the electrical activity of a population of neurons [1]. In epileptogenesis, many different factors cause or support the progression of epilepsy at different levels [2]. The epileptic seizure itself can generate an excessive amount of reactive oxidative species and reactive nitrogen species in the region of abnormal neuronal activity, and reactive oxidative species-induced changes in the structure of glutamate receptors increase neuronal excitability and seizure susceptibility [3]. In addition, oxidative stress appears to contribute to the development of neuroinflammation, neurodegeneration, and impaired neurogenesis, which in turn may promote epileptogenesis [4]; this has been confirmed in various models of epileptic rats [5,6].

Oxidative stress is defined as an imbalance between the production of cellular oxidants (reactive oxygen species such as superoxide, hydrogen peroxide, and hydroxyl radicals and lipid peroxides, as well as reactive nitrogen species such as peroxynitrite) and their removal by the following antioxidant enzymes: superoxide dismutase, catalase, glutathione peroxidase, and glutathione reductase, and antioxidants (e.g., glutathione) in the cell [7]. The antioxidant enzymes and antioxidants coordinate their activity and form an antioxidant system that reduces the oxidising agents that cause cell damage and death [8]. Peroxidation of membrane lipids, caused by an increase in free radical formation or a decrease in the activity of the antioxidant defence system, is thought to play a crucial role in controlling seizures [9].

Epilepsy is treated with oral anti-seizure medications, a chemically heterogeneous group of drugs that have been categorised into generations according to the date of their introduction into clinical use. Carbamazepine and valproate belong to the old anti-seizure medications, while lamotrigine is one of the newer drugs on the market [10]. Carbamazepine and valproate are the drugs of first choice for epilepsy in Europe and the USA. They are used to treat generalised seizures (generalised tonic-clonic seizures, absence seizures, and myoclonic seizures) and epileptic selection seizures—valproate and carbamazepine are the treatment of choice for partial seizures (simple partial, complex partial, and secondary generalized tonic-clonic seizures) and epilepsy syndromes. Valproate is the drug of choice for generalized epilepsies, and carbamazepine is the preferred drug for partial epilepsies [11]. Carbamazepine is a sodium channel blocker that affects the serotonin system [12]. The mechanisms of action of valproate are unclear, but they appear to affect GABA levels and block voltage-gated sodium channels [13]. Lamotrigine is a sodium channel-blocking antiepileptic drug that also suppresses the release of glutamate and aspartate [14]. Although they are chemically different, they all prevent seizures. However, many anti-seizure medications are metabolised to produce reactive metabolites that can bind macromolecules and thus both impair function on a molecular level and elicit systemic toxicity [15]. The role of anti-seizure medications in oxidative/antioxidative processes is controversial [4] and ranges from a moderate antioxidant to a prooxidant effect. Although the main root of its oxidant mechanisms is not scientifically established, an exacerbation of oxidative stress during antiepileptic drug therapy could also be one of the causes of pharmacotherapy-resistant epilepsy. These prooxidant effects could exacerbate seizure activity by increasing hyperexcitability and/or causing neuronal damage, which may lead to the loss of efficacy of seizure medications or apparent functional tolerance and adverse side effects. Experimental evidence suggests that almost all first-, second-, and third-generation anti-seizure medications lose their anti-seizure effect with prolonged therapy, albeit in varying degrees [16].

Human erythrocytes have effective antioxidant protection consisting of enzymatic and non-enzymatic pathways that counteract reactive oxygen species to maintain redox regulation in the body [7]. On the other hand, changes in the structure and activity of antioxidant defence enzymes under the direct influence of chronic (long-term) drug intake as biomarkers of oxidative stress in erythrocytes could be valuable parameters for the organism’s susceptibility to chronic therapy with anti-seizure medications. Interestingly, there is not much data in the literature on the effects of selected anti-seizure medications on antioxidant defence activity in the blood of treated patients. Therefore, the aim of this study was to investigate the effects of long-term monotherapy with these drugs on the antioxidant defence enzymes in the erythrocytes and oxidative (haemoglobin/methaemoglobin ratio of erythrocytes and lipid peroxide levels in plasma) and antioxidative (total glutathione and thiol groups in plasma) parameters.

## 2. Results

Before comparing the different anti-seizure therapies, the influence of the daily dosage regime on the level of selected antioxidant parameters was first measured for each of the three anti-seizure medications. Daily dosage was divided into three ranges for each anti-seizure separately. The results showed no significant differences between the daily anti-seizure medication doses in the individual groups. Therefore, our further analyses were performed on the entire group sample, regardless of the daily dosing regimen.

Compared to the control, the activity of copper-zinc superoxide dismutase was significantly reduced in the patient groups treated with lamotrigine (*p* < 0.005) and carbamazepine (*p* < 0.005). The mean value in the carbamazepine group was lower than in patients treated with lamotrigine (*p* < 0.005) and valproate (for both: *p* < 0.005), and the lamotrigine group was significantly lower compared to the valproate group (*p* < 0.01) (Figure 1).

All three anti-seizure medication therapy groups showed lower catalase activity than the control group, but there is only a statistically significant difference between the control and valproate groups (*p* < 0.005). Regarding the anti-seizure medication groups, a significant decrease in catalase activity was found between valproate and lamotrigine (*p* < 0.05) and between valproate and carbamazepine (*p* < 0.005). There were no differences in glutathione peroxidase activity in any of the four groups studied (Figure 1).

Compared to controls, there were no significant differences between glutathione reductase activities in the erythrocytes of patients treated with anti-seizure medications. However, among the groups of patients, lamotrigine had higher glutathione reductase activity than carbamazepine (*p* < 0.005), and the activity in the carbamazepine group was lower than in the group of patients treated with valproate (*p* < 0.005) (Figure 1).

No changes in haemoglobin concentration existed between the controls and the three groups of patients treated with anti-seizure medications. The average percentage concentration of oxidised haemoglobin in the erythrocytes of the participants in all groups was also not significantly different. In addition, they were very uniform (approx. 1.4% of total haemoglobin) and slightly lower in the carbamazepine patient group (Figure 1).

The concentration of total thiol groups in plasma was significantly lower in the lamotrigine group compared to the control (*p* < 0.01). There were no significant changes in total proteins or glutathione concentrations (Figure 2).

The estimated nitric oxide concentration was higher in the carbamazepine group than in the control group (*p* < 0.01). Regarding the anti-seizure medication groups, significant differences were found between lamotrigine and carbamazepine (*p* < 0.005) and between lamotrigine and valproate (*p* < 0.005) (Figure 2).

Lipid peroxides were lower in the lamotrigine group compared to the control group (*p* < 0.01) and in the carbamazepine (*p* < 0.001) and valproate (*p* < 0.001) groups (Figure 2).

For canonical discriminant analysis, we used 4 variables (antioxidant enzymes): copper-zinc superoxide dismutase, catalase, glutathione peroxidase, and glutathione reductase for each group (anti-seizure medications). Statistical analysis showed that our model was very significant (Wilks’ Lambda F (12,180) = 11.511; *p* < 0.0001) and that copper-zinc superoxide dismutase (*p* < 0.001), catalase (*p* < 0.001), and glutathione reductase (*p* < 0.001) had the largest significant differences. Chi-square tests with successive roots showed that there were two significant roots (canonical discriminant functions), D1 (*p* < 0.001) and D2 (*p* < 0.001), with their standardized coefficients for canonical variables.
D1 = −0.82 SOD1* + 0.4 CAT* + 0.14 GPx* − 0.36 GR* (*p* < 0.001) 
D2 = −0.56 SOD1* − 0.72 CAT* − 0.2 GPx* + 0.55 GR* (*p* < 0.001) * SOD1-cooper-zinc superoxide dismutase, CAT-catalase, GPx-glutathione peroxidase, and GR-glutathione reductase.

Means of canonical variables for each group (a) and the canonical scores for each case (b) are presented in Figure 3 and plotted in two-dimensional discriminant canonical space (corresponding to functions D1 and D2). The canonical discriminant analysis showed that the anti-seizure therapy led to a different composition of the antioxidant defence in the erythrocytes.

Function D1 showed that copper-zinc superoxide dismutase activity in erythrocytes contributes most to the difference. Function D2 showed that this contribution corresponds to catalase and glutathione reductase. By using the canonical variables, we were able to distinguish the control group from all the other groups and separate the lamotrigine and valproate groups from the carbamazepine group.

## 3. Discussion

In this paper, we analysed the effects of three anti-seizure monotherapies on the level of oxidative/antioxidative components in patients’ blood. Our results showed that chronic therapy changes the activity and composition of antioxidant enzymes. Long-term anti-seizure monotherapy attenuated the activity of antioxidant enzymes activity in the analysed parameters, erythrocytes, and blood plasma, but in slightly different ways. Lamotrigine therapy decreased lipid peroxides, followed by lower levels of superoxide dismutase and sulfhydryl groups in blood plasma. On the other hand, carbamazepine increased circulating nitric oxide concentration (estimated by nitrite concentrations), followed by a decrease in copper-zinc superoxide dismutase. In these two groups, the mechanisms for reducing copper-zinc superoxide dismutase appeared to differ. Although the lowest methaemoglobin levels were found in the carbamazepine group, the lower copper-zinc superoxide dismutase activity may be related to the lower pro-oxidative conditions. Methemoglobin levels in all treated groups were within the limits expected for healthy controls (1.5%), indicating that haemoglobin oxidation, as the main site for primary superoxide formation, was not increased and was even slightly reduced in the carbamazepine group. This result implies lower oxidative pressure in patients receiving long-term anti-seizure medications. This finding was even more significant in the group receiving lamotrigine, where overall lipid peroxidation decreased, as did sulfhydryl groups, indicating less oxidative stress. In addition, catalase activity was significantly reduced in the valproate group, indicating lower hydrogen peroxide levels. Since the daily dose (and thus the dosage) did not influence the patients, the changes observed were due to the long-lasting use of these medications.

From an arbitrary look, the statistical changes observed coincide with changes in other components that follow the general pattern of reduction in activity, but not uniformly. These minor changes, which were not significantly different from controls, resulted in significant differences between the groups of treated patients, which were most pronounced when glutathione reductase activity was considered; glutathione reductase activity was significantly lower in the carbamazepine group compared to valproate and lamotrigine, but not compared to controls. Therefore, the parameters indicating the composition of antioxidant enzymes considered in this study (activities of antioxidant enzymes of those considered as an antioxidant system: copper-zinc superoxide dismutase, catalase, glutathione peroxidase, and glutathione reductase) were highly significant (expressed as D1 and D2) in canonical discriminant analysis, which indicates a different composition of the antioxidant system as a whole and not as a single component. Small, non-significant changes during therapy led to a shift in antioxidant composition after a long treatment period. Thus, carbamazepine appeared to differ substantially from both valproate and lamotrigine, while all groups differed from the controls (Figure 3).

However, since anti-seizure medications are primarily considered sodium channel blockers, the difference could be due to some additional mechanisms and/or drug metabolism. Valproate is bound to albumin and affects the GABA-ergic and endocrine systems, whereas carbamazepine needs to be converted to epoxide via cytochrome P450 to be bioactive, increasing reactive oxygen species. All three substances are metabolised and excreted by the formation of glucuronide conjugates. Valproate has been reported to reduce glutathione and superoxide dismutase levels in the brain of control rats after 45 days, demonstrating that anti-seizure medications alter the levels of oxidants/antioxidants per se [17]. The same study showed that lamotrigine did not induce significant changes in oxidative/antioxidative balance parameters in the brains of control rats. Considering all these processes, long-term therapy with anti-seizure medications is a factor that affects the balance between oxidants and antioxidants and, thus, the antioxidant defence in different ways, which in turn is reflected in the blood oxidative/antioxidative balance of the blood.

This phenomenon was shown in a study on untreated epileptic children and children receiving monotherapy with valproic acid and carbamazepine for up to 7 and 12 months, respectively [9]. However, some previous results on the effects of anti-seizure medications on antioxidant activity during therapy are contradictory, ranging from a moderate antioxidant effect to an enhancement of the oxidative process [4]. Our results showed that anti-seizure medications reduce antioxidant enzymes, suggesting lower oxidative stress, as shown in previous studies, but various antioxidant parameters were lower [3,18]. The protective effect likely depends on the therapy duration and sampling time. Differences in the design of similar studies should be taken into account, mainly regarding the age of the subjects, the period (a few years or more than ten years) during which they have been taking prescribed medication, the daily doses of these drugs, and whether the control group consisted of healthy subjects or naïve neurological patients.

In one study that analysed five enzymes (including glutathione-S-transferase), the activity of glutathione peroxidase was significantly lower, and the activity of glutathione reductase was considerably higher in patients treated with valproate [19]. Our results also showed an involvement of glutathione reductase activity, which was not significantly different compared to controls, but was significantly different between patients on anti-seizure medications. Consistent with our results, some other authors also found that valproate therapy did not change glutathione peroxidase activity [20]. Prolonged valproate therapy (7–14 years) significantly increased copper-zinc superoxide dismutase activity, contradicting our results. Still, the control group in this study consisted of clinically healthy individuals and not patients [21]. Peker et al. (2009) found that valproate therapy increased serum levels of nitric oxide in epileptic children by about 10% compared to the control group [22]. Karabiber et al. (2004) also showed that nitrite and nitrate levels were significantly higher in epileptic children treated with valproate and carbamazepine [23]. Geronzi et al. (2018) showed that carbamazepine increased the release of ATP and nitric oxide-derived metabolites from erythrocytes into the lumen, leading to an increased nitric oxide pool in the vasculature [24]. Our results confirmed this increase of elevated nitric oxide in the blood serum of patients from the carbamazepine group, which appeared to be one of the common carbamazepine effects.

Cengiz et al. (2000) found a significant decrease in total glutathione concentration in the valproate and carbamazepine groups [25]. No significant differences were found in our studies. However, a notable difference in our study design is that the subjects were children aged 1–15 years with epilepsy and healthy controls (without epilepsy). This finding also suggests that the oxidative/antioxidative effects of anti-seizure medications depend on the age of the patients and cannot be generalised, with important considerations for the interaction(s) with other medications, age, and morbidity.

With regard to lamotrigine, Huang et al. (2014) discovered a significant decrease in copper-zinc superoxide dismutase activity, similar to our results, and observed a decrease in glutathione peroxidase after at least 36 months of monotherapy with this drug [26]. They also found that total glutathione concentrations in blood plasma were much higher in treated patients than the control subjects, with no changes in non-glutathione sulfhydryl compounds (thiols) and no differences in lipid peroxide concentrations between groups. Other studies investigating lamotrigine have shown lower lipid peroxidation than untreated epilepsy patients or healthy controls, e.g., in [27]. The same result was also obtained in our study.

Our results showed that the mechanisms of antioxidant action of the studied anti-seizure medications differ, leading to a shift in antioxidant composition reflecting the oxidant-antioxidant balance, which is considered an essential process in epileptogenesis. Although our study has some limitations (e.g., the comparison between drug-naive patients and patients receiving long-term monotherapy, excluding a group of healthy individuals chronically treated with these drugs and the comparison between healthy individuals without medication and healthy individuals with medication and epileptics with medication), we have shown that long-term therapy (at least six months) with the anti-seizure medications studied alters the balance between oxidants and antioxidants through different mechanisms and components. The exact impact of the observed changes on pathophysiology cannot be determined without parallel studies of clinical manifestations in patients.

However, our study was conducted when the epileptogenic period was over, therapy was underway, clinical manifestations were in the foreground, and the antioxidant system was tracking not only the reactive oxygen species involved in epilepsy but also the effects of medications at different levels. Studying antioxidant systems can help assess the role, importance, and extent of particular anti-seizure medications in regulating the balance between oxidants and antioxidants. This approach may help to establish antioxidant protocols for complementary effects, including oxidative stress in epileptogenesis and epilepsy. Several antioxidants are considered protective in epilepsy [5], such as *α*-tocopherol [28], coenzyme Q_10_ [6], melatonin [29], resveratrol [30], α-lipoic acid [31], curcumin [32], ascorbic acid [33], and vitamin D [34], but none of them showed a large and significant effect. Therefore, the combination of anti-seizure medications and antioxidants could be helpful. Still, the combination must be carefully considered since each anti-seizure medication affects the balance between oxidants and antioxidants through its specific mechanism.

## 4. Materials and Methods

### 4.1. Subjects

This study was conducted at the Institute of Mental Health Belgrade, Serbia, in 2019 and 2020 in accordance with the Declaration of Helsinki and the Good Clinical Practice guidelines. The study protocol was approved by the Ethics Committee of the Institute of Mental Health School of Medicine, University of Belgrade, Belgrade, Serbia (licence No. 30/26), and patients gave informed written consent.

Patients with epilepsy aged 18–70 years were recruited consecutively during their regular appointments at the Department of Epilepsy and Clinical Neurophysiology of the Institute of Mental Health in Belgrade. The diagnosis and type of epilepsy were established by a neurologist (M.M.) [35,36] based on seizure semiology, neurological examination, and video-electroencephalography (V-EEG). Of the final sample of 67 patients with epilepsy, 24 had idiopathic generalised epilepsy, and 43 had focal epilepsy. Based on more than six months of monotherapy with anti-seizure medications (lamotrigine, carbamazepine, and valproate), three groups were formed with 22 and 23 epilepsy patients of both sexes and different ages (Table 1). In the semi-structured interview, each patient’s demographic and clinical characteristics, the dosage and administration regimen of the seizure medication, concomitant diseases, and drug therapy were recorded. Patients were initially categorised into three ranges for each antiepileptic drug according to the daily dosage (lamotrigine group: <100 mg, 100–200 mg, and >200 mg per day; valproate group: <500 mg, 500–1000 mg, and >1000 mg per day; carbamazepine <400 mg, 400–800 mg, and >800 mg per day). As there were no significant differences in the measured parameters due to daily dosing, they were further summarised as one group for each anti-seizure medication. An exclusion criterion was the presence of a severe concomitant disease that required the concomitant use of other medications. The concomitant diseases of the patients studied ranged from hypovitaminosis D, hypertension, and hyperlipidaemia to fibrillatio atriorum, chorioretinitis focalis, and posthysterectomy. Other medications besides anti-seizure medications included vitamin D, captopril, propafenone, acetylsalicylic acid, and hydrochlorothiazide. The control group consisted of drug-free epilepsy patients before the start of seizure therapy (to compare the results obtained). Comorbidities in the control group mainly included headaches and, in one case, diabetes.

### 4.2. Blood Sampling Collection, Isolation and Haemolysis of Erythrocytes

Venous blood (approximately 5 mL) was obtained after overnight fasting (10–12 h) and stored directly in vacutainers coated with an anticoagulant film (EDTA, 1 g/L). Within 48 h after sampling, the whole blood sample was processed. The erythrocytes were settled from the blood via centrifugation at 3000× *g* for 10 min. The plasma in the supernatant was separated and stored in the freezer. The intermediate layer was discarded, and the erythrocytes were washed three times with five volumes of saline (0.9% NaCl) and separated via centrifugation, as described above. Erythrocyte hemolysis was performed by adding 3 mL of ice-cold MiliQ water to 0.5 mL of washed erythrocytes, followed by mixing. The obtained haemolysates were stored in the freezer until use (maximum seven days).

### 4.3. Preparation of Haemolyses for Determination of Antioxidant Enzyme Activity

To determine the activity of copper-zinc superoxide dismutase, it was necessary to first precipitate the haemoglobin from the analysed haemolyses [37]. Briefly, 0.4 mL of ice-cold distilled water was added to 0.4 mL of haemolysate (in 2 mL Eppendorf tube), followed by 0.4 mL of ice-cold ethanol. After stirring the samples briefly on a Vortex, 0.3 mL of ice-cold chloroform was added dropwise to the resulting suspension. After vigorous stirring on a vortex (about 30 s per sample), the contents were centrifuged (5000× *g*, 5 min). The resultant clear (and colourless) supernatants were separated from the precipitated proteins (haemoglobin). These samples were left in the freezer until determination. The adrenaline method was used to determine total copper-zinc superoxide dismutase activity [38]. One copper-zinc superoxide dismutase unit was defined as the amount of the enzyme necessary to decrease the rate of adrenaline auto-oxidation by 50% at pH 10.2. Copper-zinc superoxide dismutase activity was expressed in activity units per gram of haemoglobin (U/g Hb).

Aliquots of haemolysate were used to determine the activity of catalase, glutathione peroxidase, and glutathione reductase using a Shimadzu UV-160 spectrophotometer (Tokyo, Japan).

This haemolysate was further diluted to determine catalase activity: 10 µL of haemolysate, 10 µL of 96% ethanol, and 980 µL of distilled water were added, and the resulting solution was mixed well on a Vortex. Catalase activity in dilute haemolysate was determined according to [39]. One unit of catalase activity was defined as the amount of the enzyme that decomposes 1 mmol H_2_O_2_ per minute at 25 °C and pH 8.0; catalase activity was expressed in activity units per gram of haemoglobin (U/g Hb).

Selenium-dependent glutathione peroxidase activity was determined by the glutathione reduction of t-butyl hydroperoxide using a modification of the assay described by Paglia and Valentine (1967) [40]. Glutathione peroxidase activity is expressed as the amount of the enzyme needed to oxidise 1 μmol NADPH per minute at 25 °C and pH 7.0 per gram of haemoglobin; the unit of activity is µM NADPH min^−1^ g Hb^−1^. Glutathione reductase activity was determined using the method of Glatzle and colleagues (1974) [41]. One unit of glutathione reductase activity was defined as the amount of the enzyme needed to oxidise 1 μmol NADPH per minute at 25 °C and pH 7.4 per gram of haemoglobin.

### 4.4. Determination of Total and Methaemoglobin in Haemolyses

The total haemoglobin content in the analysed haemolyses was determined by the standard Drabkin cyano-methaemoglobin method [42]. The absorbance was measured at 545 nm (Shimadzu UV-1800 UV/Visible Scanning Spectrophotometer 1800; Kyoto, Japan). The haemoglobin concentration was expressed in mg/mL. The content of oxidised methaemoglobin was determined based on the unique spectral characteristics of this haemoglobin derivative: a small peak at 630 nm [43]. To record the spectra, we used a Thermo Scientific NanoDrop 2000c spectrophotometer, Waltham, USA. Levels of methaemoglobin are expressed in percentages of total haemoglobin.

### 4.5. Determination of Plasma Parameters

Total protein concentration, concentration of total plasma thiols, nitrite concentration, lipid peroxides level, and total glutathione level were determined in the stored plasma (after centrifugation of whole blood). Total protein concentration was measured with an accurate and rapid method with Biuret reagent [44]. The addition of a cupric salt to an alkaline protein solution produces a reddish-violet colour, measuring 546 nm. The total protein concentration was expressed in grams per litre (g/L). A spectrophotometric assay based on 5,5′-dithiobis(2-nitrobenzoic acid) acid (DTNB or Ellman’s reagent) was used to determine total plasma thiols [45]. Concentrations were expressed in mM. Nitric oxide concentration was indirectly measured by reconversion of nitrate/nitrite to nitric oxide by Griess reagent [46]. The method was based on the formation of a coloured azo compound with maximum absorption at 548 nm in the reaction between sulfanilic acid (or sulfanilamide) and *N*-(1-naphthyl)ethylenediamine in an acidic environment in the presence of nitrite. Nitrite concentration was expressed in µM. Total protein concentration, total plasma thiols, and nitrite concentration were measured using a Shimadzu UV-1800 spectrophotometer (Tokyo, Japan).

The level of lipid peroxides was determined using a Biotek Synergy H1 microplate reader, according to the method described by [47]. In this method, we used a thiobarbituric acid assay to measure lipid peroxide levels in the plasma. Lipids and proteins were precipitated using a phosphotungstic acid-sulfuric acid system to eliminate disturbing substances such as glucose and water-soluble aldehydes. Also, the reaction of thiobarbituric acid with these lipids was carried out in an acetic acid solution to avoid its reaction with other substances, such as sialic acid. After the thiobarbituric acid reaction, the product was measured via fluorometry because the thiobarbituric acid reaction product is a fluorescent substance. Levels of lipid peroxides are expressed in nmol per ml of blood. The total glutathione concentration was determined on the same reader, according to the method described by [48]; the concentration was expressed in g/L.

### 4.6. Statistical Analyses

Statistical analyses were performed according to the protocols described by Hinkle et al. (2002), Manley (1986), and Blagojević et al. (1998) [49,50,51,52], which have been subsequently applied to study neurological conditions [53,54,55]. The results are expressed as the mean value followed by the standard error of the mean (SEM). The data were analyzed using analysis of variance (ANOVA) and Tukey post-hoc Honest Significant Difference (HSD) test for unequal sample size was used to determine significant differences between groups. The canonical discriminant analysis measured the effects of anti-seizures on overall connections between individual antioxidant enzyme components at the total group level and the significance of these changes. Correlation analysis was used to calculate the relations between individual components, but an alternative statistical test, canonical discriminant analysis, was used to calculate differences between groups, considering the complete correlation matrix of a particular group (i.e., the composition of antioxidant defence enzymes) to others. A probability level of *p* < 0.05 was considered statistically significant.

## 5. Conclusions

Our results suggest that long-term monotherapy with these anti-seizure medications can affect oxidant/antioxidant balance, where copper-zinc superoxide dismutase (lamotrigine and carbamazepine) and catalase (valproate) were primarily affected (decreased). Lamotrigine significantly reduced the level of lipid peroxides and the total plasma concentration of the sulfhydryl group, suggesting lower oxidative stress and protectivity. In addition, carbamazepine increased the formation of nitric oxide, which seems to be a common effect. We demonstrated that long-term therapy with anti-seizure medications can change the oxidant/antioxidant balance through different mechanisms and components.

## Figures and Tables

**Figure 1 pharmaceuticals-17-00130-f001:**
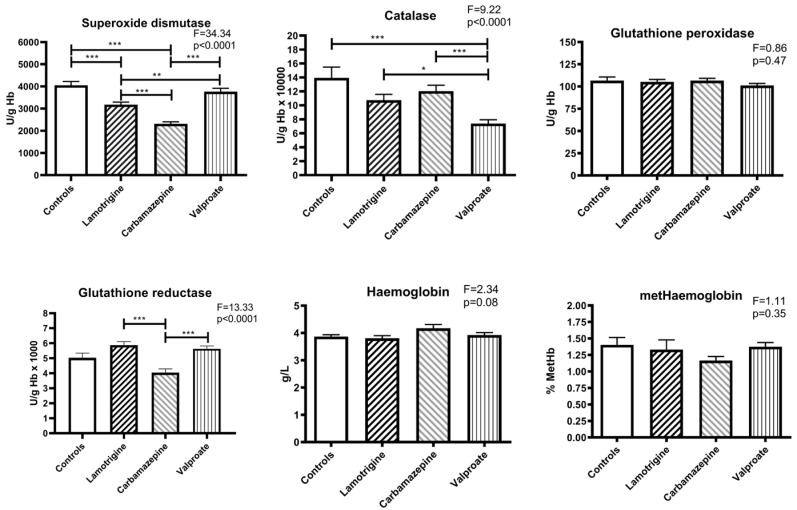
The activity of antioxidant enzymes and concentration of haemoglobin and methaemoglobin in erythrocytes of patients treated with anti-seizure medications and the control group. Statistically significant differences between the experimental and control groups: * *p* < 0.05, ** *p* < 0.01, and *** *p* < 0.001.

**Figure 2 pharmaceuticals-17-00130-f002:**
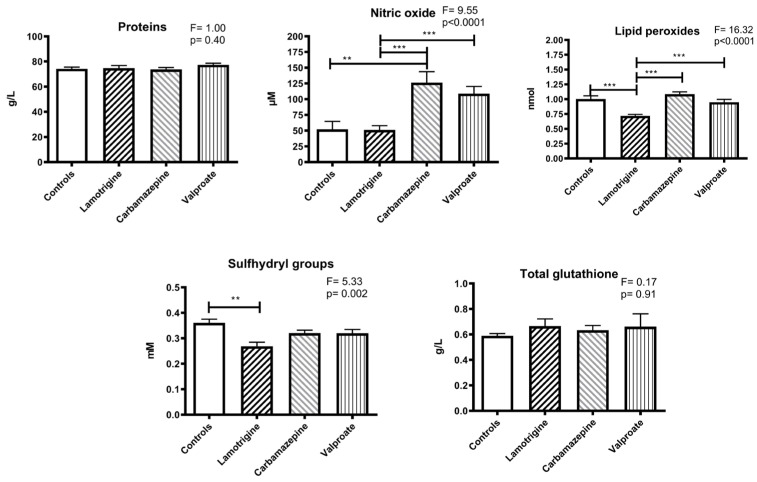
The levels of total proteins, lipid peroxides, sulfhydryl groups, glutathione, and nitric oxide (as nitrite) in the blood plasma of patients treated with anti-seizure medications and the control group. Statistically significant differences between the experimental and control groups: ** *p* < 0.01, and *** *p* < 0.001.

**Figure 3 pharmaceuticals-17-00130-f003:**
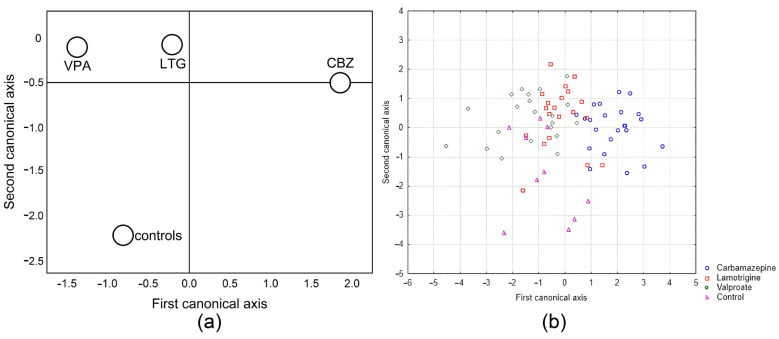
Two-dimensional discriminant analysis of patients treated with anti-seizure and the control group (**a**) and the canonical score for each case (**b**).

**Table 1 pharmaceuticals-17-00130-t001:** Anamnestic data for patients who participated in this study.

Group	Gender		Average Years (Mean ± SD)	Smokers	Other Medications	Comorbidity
	M	F				
Controls	4	6	44.9 ± 18.4	1/10	3/10	4/10
Lamotrigine	7	15	64.2 ± 17.9	0/22	15/22	18/22
Carbamazepine	12	11	42.1 ± 16.5	9/23	15/23	19/23
Valproate	19	3	33.6 ± 19.7	5/22	5/22	6/22

## Data Availability

Data is contained within the paper.

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
