# Peer review of "Influence of Long-Term Anti-Seizure Medications on Redox Parameters in Human Blood"

_pharmaceuticals, 2024, doi:10.3390/ph17010130_

Round 1

Reviewer 1 Report

Comments and Suggestions for Authors

Authors still used the term “anti-epileptic drugs” instead of “antiseizure medication”. Please use the term “antiseizure medication” (ASM). 

Why did the authors only select patients who used lamotrigine, carbamazepine, and valproate? Authors should elaborate on the scientific reason in the background. 

There is a duplication paragraph in the results section (lines 90-95) and the method section (lines 285-291). 

What did “D1” (line 136) and “D2” (line 137) refer to?

What suggestion would the author give based on this study? Would it be beneficial to give anti-oxidants to epilepsy patients who are treated with these drugs?

Author Response

We are thankful for the reviewers' constructive comments that have helped us significantly improve the manuscript and clarify questioned issues.

Point-by-point answers (A) to all four reviewers' comments and questions (Q) are provided separately in one file.

(Q) Authors still used the term “anti-epileptic drugs” instead of “anti-seizure medication”. Please use the term “anti-seizure medication” (ASM).

(A) The reviewer is correct; we apologise. This significant change has been made in this version of the text. Thank you.

(Q) Why did the authors only select patients who used lamotrigine, carbamazepine, and valproate? Authors should elaborate on the scientific reason in the background. 

(A) Thank you for the suggestion; we have added the paragraph in the background. Also, lamotrigine, carbamazepine, and valproate are the most commonly used anti-seizure medications in our practice for epilepsy patients.

(Q) There is a duplication paragraph in the results section (lines 90-95) and the method section (lines 285-291). 

(A) We removed the duplicate paragraph from the results section. Thank you.

(Q) What did “D1” (line 136) and “D2” (line 137) refer to?

(A) Thanks for the comment! We omitted by mistake the paragraph from the previous manuscript version that described D1 and D2 functions! Now that information is added.

(Q) What suggestion would the author give based on this study? Would it be beneficial to give anti-oxidants to epilepsy patients who are treated with these drugs?

It was written in the last paragraph of the Discussion section. Anyway, we have added one more sentence in the Conclusions. Thank you.

Reviewer 2 Report

Comments and Suggestions for Authors

A few comments are listed below.

Change antiepileptic drugs, antiepileptic therapies, antiepileptic, antiepileptic drug therapy, antiepileptic monotherapy, by antiseizure drugs, throughout the entire manuscript.

Line 202. …. [borowitcg]. What it means?

Materials and Methods:

How were epileptic patients diagnosed? What type of epilepsy was diagnosed? Did they present focal or generalized seizures? This information should be included.

Control group: drug-naive patients. It is not clear if these patients are epileptic or suffer from another neurological disease that requires the use of antiseizure drugs. How were these patients recruited? Were they patients with a recent diagnosis of epilepsy or were they patients who had their medication discontinued? This information should be included.

Which are the comorbidities identified and indicated in Table 1? Indicate it.

What were the other medications indicated in Table 1? Indicate it.

Comments on the Quality of English Language

 Minor editing of English language required.

Author Response

We are thankful for the reviewers' constructive comments that have helped us significantly improve the manuscript and clarify questioned issues.

Point-by-point answers (A) to all four reviewers' comments and questions (Q) are provided separately in one file.

(Q) Authors still used the term “anti-epileptic drugs” instead of “anti-seizure medication”. Please use the term “anti-seizure medication” (ASM).

(A) The reviewer is correct; we apologise. This significant change has been made in this version of the text. Thank you.

(Q) Why did the authors only select patients who used lamotrigine, carbamazepine, and valproate? Authors should elaborate on the scientific reason in the background. 

(A) Thank you for the suggestion; we have added the paragraph in the background. Also, lamotrigine, carbamazepine, and valproate are the most commonly used anti-seizure medications in our practice for epilepsy patients.

(Q) There is a duplication paragraph in the results section (lines 90-95) and the method section (lines 285-291). 

(A) We removed the duplicate paragraph from the results section. Thank you.

(Q) What did “D1” (line 136) and “D2” (line 137) refer to?

(A) Thanks for the comment! We omitted by mistake the paragraph from the previous manuscript version that described D1 and D2 functions! Now that information is added.

(Q) What suggestion would the author give based on this study? Would it be beneficial to give anti-oxidants to epilepsy patients who are treated with these drugs?

It was written in the last paragraph of the Discussion section. Anyway, we have added one more sentence in the Conclusions. Thank you.

(Q) Change antiepileptic drugs, antiepileptic therapies, antiepileptic, antiepileptic drug therapy, antiepileptic monotherapy, by anti-seizure drugs, throughout the entire manuscript.

(A) Criticism accepted. We have finally done it! Thank you.

(Q) Line 202. …. [borowitcg]. What it means?

(A) Thank you for noticing this omission. We missed entering the reference, and now we have added it. It is the reference No. [4].

(Q) Materials and Methods: (1) How were epileptic patients diagnosed? What type of epilepsy was diagnosed? Did they present focal or generalized seizures? This information should be included. (2) Control group: drug-naive patients. It is not clear if these patients are epileptic or suffer from another neurological disease that requires the use of anti-seizure drugs. How were these patients recruited? Were they patients with a recent diagnosis of epilepsy or were they patients who had their medication discontinued? This information should be included. Which are the comorbidities identified and indicated in Table 1? Indicate it. What were the other medications indicated in Table 1? Indicate it.

(A) All above-mentioned criticisms accepted; thank you. In the section Material and Methods in the revised text, this parts of section 4.1 (Subjects) now reads:

"Patients with epilepsy aged 18-70 years were recruited consecutively during their regular appointments at the Department of Epilepsy and Clinical Neurophysiology of the Institute of Mental Health in Belgrade. The diagnosis and type of epilepsy were established by a neurologist (M.M.) [37,38] based on seizure semiology, neurological examination and video-electroencephalography (V-EEG). Of the final sample of 67 patients with epilepsy, 24 had idiopathic generalised epilepsy, and 43 had focal epilepsy. Based on more than six months of monotherapy with anti-seizure medications (lamotrigine, carbamazepine, valproate), three groups were formed with 22 and 23 epilepsy patients of both sexes, different ages (Table 1). In the semi-structured interview, each patient's de-mographic and clinical characteristics, the dosage and administration regimen of the seizure medication, concomitant diseases and drug therapy were recorded. Patients were initially categorised into three ranges for each antiepileptic drug according to daily dosage (lamotrigine group <100 mg, 100-200 mg, and >200 mg per day; valproate <500 mg, 500–1000 mg, >1000 mg per day; carbamazepine <400 mg, 400–800 mg, >800 mg per day). As there were no significant differences in the measured parameters due to daily dosing, they were further summarised as one group for each anti-seizure medication. An exclusion criterion was the presence of a severe concomitant disease that required the concomitant use of other medications. The concomitant diseases of the patients studied ranged from hypovitaminosis D, hypertension, hyperlipidaemia to fibrillatio atriorum, chorioretinitis focalis and posthysterectomy. Other medications besides anti-seizure medications included vitamin D, captopril, propafenone, acetylsalicylic acid and hydrochlorothiazide. The control group consisted of drug-free epilepsy patients before the start of seizure therapy (to compare the results obtained). Comorbidities in the control group mainly included headaches and, in one case, diabetes."

Reviewer 3 Report

Comments and Suggestions for Authors

Lines 136-141: Authors wrote: “From the above functions, it can be concluded from D1 that copper-zinc superoxide dismutase activity in erythrocytes contributes most to the difference…” But there is no any information above about the functions.

Besides, in Discussion, Fig. 3 “Two-dimensional discriminant analysis…”, but I think the figure include final results of the analysis only. If authors want to show such analysis, they should show main steps of this analysis, including primary data for each patient (see, for example Fig. in the paper by Korth M. et al.; doi: 10.1007/BF00920218).

Minor comments:

Please use common abbreviations for some terms, like superoxide dismutase etc. in Abstract and the manuscript.

Line 51: Where is ]?

Author Response

We are thankful for the reviewers' constructive comments that have helped us significantly improve the manuscript and clarify questioned issues.

Point-by-point answers (A) to all four reviewers' comments and questions (Q) are provided separately in one file.

(Q) Lines 136-141: Authors wrote: “From the above functions, it can be concluded from D1 that copper-zinc superoxide dismutase activity in erythrocytes contributes most to the difference…” But there is no any information above about the functions.

(A) Thanks for the comment! We omitted by mistake the paragraph from the previous manuscript version that described D1 and D2 functions! Now that has been fixed.

(Q) Besides, in Discussion, Fig. 3 “Two-dimensional discriminant analysis…”, but I think the figure include final results of the analysis only. If authors want to show such analysis, they should show main steps of this analysis, including primary data for each patient (see, for example Fig. in the paper by Korth M. et al.; doi: 10.1007/BF00920218).

Thanks for the comment. Yes, we included the final results of the analysis in a way that does not confuse the reader with too many statistics. Now, we have added the following part in the Results section:

"For canonical discriminant analysis, we used 4 variables (antioxidant enzymes): copper-zinc superoxide dismutase, catalase, glutathione peroxidase and glutathione reductase for each group (anti-seizure medications). Statistical analysis showed that the used model was highly statistically significant (Wilks' Lambda F (12,180) = 11.511; p<0.0001) and that copper-zinc superoxide dismutase (p<0.001), catalase (p<0.001) and glutathione reductase (p<0.001) confirmed to difference significantly the most. Chi-square tests with successive roots showed that there were two significant roots (canonical discriminant functions), D1 (p<0.001) and D2 (p<0.001), with their standardized coefficients for canonical variables.

D1 = -0.82 SOD1 + 0.4 CAT + 0.14 GPx – 0.36 GR (p<0.001)*

D2 = -0.56 SOD1 – 0.72 CAT – 0.2 GPx + 0.55 GR (p<0.001)*

*SOD1-cooper-zinc superoxide dismutase, CAT-catalase, GPx-glutathione peroxidase, GR-glutathione reductase.

Means of canonical variables were presented in Figure 3 for each group (a) as well as canonical scores for each case (b) and plotted in two-dimensional discriminant canonical space (corresponding to functions D1 and D2). The canonical discriminant analysis showed that the anti-seizure therapy investigated led to a different composition of the antioxidant defence in the erythrocytes."

Minor comments:

(Q) Please use common abbreviations for some terms, like superoxide dismutase etc. in Abstract and the manuscript.

As the second reviewer asked us to avoid abbreviations, therefore we have removed all abbreviations from the text.

(Q) Line 51: Where is ]?

Thank you for noticing, we changed it in the revised text.

Reviewer 4 Report

Comments and Suggestions for Authors

Dear Authors,

I take note of the changes you have made to the manuscript. There are – as far as I'm concerned – no other critical issues to report.

Best regards.

Author Response

We thank the reviewer for his kind words and his efforts to make the manuscript look better

Round 2

Reviewer 3 Report

Comments and Suggestions for Authors

The paper could be accepted in the present form.